# Effect of Fibroblast Growth Factor 21 on the Development of Atheromatous Plaque and Lipid Metabolic Profiles in an Atherosclerosis-Prone Mouse Model

**DOI:** 10.3390/ijms21186836

**Published:** 2020-09-17

**Authors:** Hyo Jin Maeng, Gha Young Lee, Jae Hyun Bae, Soo Lim

**Affiliations:** 1Department of Internal Medicine, Seoul National University College of Medicine, Seoul National University Bundang Hospital, Seongnam 13620, Korea; mhj5398@naver.com (H.J.M.); ghayoung@gmail.com (G.Y.L.); 2Department of Internal Medicine, Korea University Anam Hospital, Korea University College of Medicine, Seoul 02841, Korea; fermatah@gmail.com

**Keywords:** fibroblast growth factor 21, atherosclerosis, diabetes mellitus, inflammation

## Abstract

Fibroblast growth factor 21 (FGF21) is a hormonal regulator of lipid and glucose metabolism. We aimed to investigate the effect of an FGF21 analogue (LY2405319) on the development of atherosclerosis and its associated parameters. ApoE^−/−^ mice were fed an atherogenic diet for 14 weeks and were randomly assigned to control (saline) or FGF21 (0.1 mg/kg) treatment group (*n* = 10/group) for 5 weeks. Plaque size in the aortic arch/valve areas and cardiovascular risk markers were evaluated in blood and tissues. The effects of FGF21 on various atherogenesis-related pathways were also assessed. Atherosclerotic plaque areas in the aortic arch/valve were significantly smaller in the FGF21 group than in controls after treatment. FGF21 significantly decreased body weight and glucose concentrations, and increased circulating adiponectin levels. FGF21 treatment alleviated insulin resistance and decreased circulating concentrations of triglycerides, which were significantly correlated with plaque size. FGF21 treatment reduced lipid droplets in the liver and decreased fat cell size and inflammatory cell infiltration in the abdominal visceral fat compared with the control group. The monocyte chemoattractant protein-1 levels were decreased and β-hydroxybutyrate levels were increased by FGF21 treatment. Uncoupling protein 1 expression in subcutaneous fat was greater and fat cell size in brown fat was smaller in the FGF21 group compared with controls. Administration of FGF21 showed anti-atherosclerotic effects in atherosclerosis-prone mice and exerted beneficial effects on critical atherosclerosis pathways. Improvements in inflammation and insulin resistance seem to be mechanisms involved in the mitigation of atherosclerosis by FGF21 therapy.

## 1. Introduction

Diabetes mellitus (DM) is a noncommunicable disorder that threatens human health because of its various complications. As of 2019, an estimated 463 million people worldwide suffered from DM. The number of individuals with DM is expected to continue to grow, reaching about 700 million by 2045 (International Diabetes Federation, Diabetes Atlas 9th edition). If not properly treated, DM is accompanied by multiple complications in the eyes, kidneys, nerves, coronary arteries, and cerebrovascular and peripheral blood vessels, resulting in significant illness and incurring huge costs medically and socioeconomically [1].

There are many antidiabetic medications available at present and some have proven beneficial effects on the cardiovascular system as well as giving glycemic control. However, they have several side effects, such as genital infection with sodium-glucose cotransporter-2 (SGLT-2) inhibitors, and gastrointestinal discomfort with glucagon like peptide-1 (GLP-1) analogues. New compounds have been developed to overcome the side effects of existing drugs. Among several candidates used for this purpose, fibroblast growth factor 21 (FGF21) has drawn much attention. This is a hormone produced mainly by the liver [2]. Physiologically, the production and secretion of FGF21 increase in response to stress in the body, acting as an adaptive response [3]. Administration of FGF21 for pharmacological purposes shows favorable multifaceted effects on metabolically important organs [4].

The mechanism of action of FGF21 and the tissues responsible for its effects have not yet been defined: both adipose tissues and the central nervous system have been identified as the targets mediating FGF21-dependent increases in insulin sensitivity, energy expenditure, and weight loss. A recent study suggested that, while FGF21 signaling to adipose tissue is required for the acute insulin-sensitizing effects of FGF21, such signaling is not required for its chronic effects in increasing energy expenditure and lowering body weight [5]. That study also showed that FGF21 acutely enhanced insulin sensitivity through actions on brown adipose tissue (BAT). Thus, the effects of FGF21 can be mediated by a variety of tissues.

A study with apolipoprotein (Apo) E^–/–^ mice showed that administration of FGF21 inhibited decreased neointima formation in the carotid artery and macrophage-mediated inflammation in the aortic sinuses and brachiocephalic arteries [6]. In another study, administration of an FGF21 analogue (LY2405319) for 1 month in humans with DM exhibited many favorable metabolic effects, such as weight loss and improvements in lipid profiles [7]. However, its effects on atherosclerosis and related pathways were not revealed fully. Here, we investigated the effects of FGF21 on cardiometabolic parameters in atherosclerosis-prone ApoE^−/−^ mice. Variables related to glucose homeostasis, lipid metabolism, and the development and progression of atherosclerosis were examined in this rodent model.

## 2. Results

### 2.1. Changes in Weight during FGF21 Treatment

Weight changes were compared between the control and FGF21 groups for the 5 weeks of treatment (from 29 weeks of age). The mean weight of mice in the FGF21 group began to fall from the beginning compared with that in the control group, and the difference was maintained thereafter (Figure 1A). Food intake did not differ between the groups (Appendix A).

### 2.2. Intraperitoneal Glucose Tolerance Test (IPGTT) after Treatment

Glucose concentrations were lower in the FGF21-treated mice compared with controls during the 120 min IPGTT study at the end of 5 weeks of treatment (Figure 1B). The area under the curve of glucose (AUC_glucose_) after the IPGTT was also lower in the FGF21-treated mice compared with the controls.

### 2.3. Biochemical Variables after FGF21 Treatment

In the ApoE^−/−^ mice, fasting glucose and insulin levels were lower in the FGF21 treatment group than in the control group (Figure 1C,D). Consequently, the HOMA-IR value, a surrogate marker of insulin resistance, was lower in the FGF21-treated group than in the control group, which was consistent after being adjusted for final body weight (Figure 1E).

The HDL-cholesterol levels were significantly higher in the FGF21 group compared with the control group (Figure 2).

Levels of triglycerides and LDL/VLDL cholesterol were lower in the FGF21 group compared with the control group with borderline significance. Adiponectin and β-hydroxybutyrate levels were significantly higher and glucagon levels were significantly lower in the FGF21 group than in the control group (Figure 3). MCP-1 levels were lower in the FGF21 group than in the control group, but this did not reach statistical significance. There was no difference in the resistin levels between the two groups (Figure 3). The circulating concentrations of TNF-α and IL-6 did not differ significantly between groups.

### 2.4. Atheroma Burden in the Aorta of ApoE^−/−^ Mice

Plaque accumulation in the aortic arch of ApoE^−/−^ mice is shown in Figure 4A. The plaque area was significantly lower in the FGF21 group compared with the control group after adjusting for body weight (Figure 4B).

The atheroma burden in the aortic valve area was also significantly lower in the FGF21-treated mice than in the control mice (Figure 4C,D).

### 2.5. Adipocyte Size and Crown-Like Structures in Abdominal Visceral Adipose Tissue

After 5 weeks of treatment, crown-like structures, indicating infiltration of inflammatory cells, were found more frequently in the control group but not in the FGF21 group (Figure 5A). Adipocytes in abdominal visceral adipose tissue of mice in the FGF21 group were smaller than those in the control group (Figure 5B).

### 2.6. Immunofluorescence Staining of CD68 in Abdominal Visceral Adipose Tissue

Immunofluorescence staining for inflammatory cells in the abdominal visceral adipose tissue showed that FGF21 treatment reduced the immunopositivity levels of CD68 significantly compared with the controls (Figure 5C,D).

### 2.7. Adipocyte Size and UCP1 Positivity in Abdominal Subcutaneous Adipose Tissue

Adipocytes in subcutaneous adipose tissue of mice in the FGF21 group were smaller than in the control group (Figure 6A). Immunostaining for UCP1 showed that FGF21 treatment increased the immunopositivity levels of UCP1 compared with the controls (Figure 6B).

### 2.8. Adipocytes Size in BAT

Adipocytes in the BAT of FGF21-treated mice were smaller than those in the control group (Figure 6C,D). Multilocular lipid droplets were also found in the BAT of FGF21-treated mice.

### 2.9. Area and Size of Lipid Droplets in the Liver

After 5 weeks of treatment, the livers of mice in the FGF21 group displayed fewer fatty changes compared with the control group (Figure 7A). The lipid droplet areas (Figure 7B) in the liver of FGF21-treated mice were smaller than those in the control group.

### 2.10. Safety of FGF21 Therapy

The mice treated with FGF21 showed no sign of significant side effects, such as edema.

## 3. Discussion

This study demonstrated that treatment with FGF21 reduced atheromatous plaque formation in the aortic arch and valves, using an atherosclerosis-prone mouse model. Body weights decreased and dynamic glycemic excursion after the IPGTT was also lower in the FGF21 group than in the control group. The FGF21 therapy decreased fasting insulin levels and HOMA-IR, suggesting an improvement in insulin resistance. Lipid profiles were improved, and adiponectin levels were increased by FGF21 treatment, suggesting that it has a positive effect on insulin sensitivity and that it has an anti-inflammatory action.

Administration of FGF21 has been reported to improve atherosclerosis risk factors, including blood glucose concentrations, blood lipids, and weight in animal models using rodents and primates [8,9]. Several studies have been conducted to investigate the role of FGF21 in atherosclerosis [6,7,10,11,12]. A study using ApoE^−/−^ mice reported that the administration of FGF21 alleviated atherosclerosis by ameliorating Fas-mediated apoptosis [12]. Another study demonstrated the protective effect of FGF21 on the proliferation and migration of vascular smooth muscle cells via inhibition of the nucleotide-binding domain leucine-rich repeat and pyrin domain containing receptor 3 (NLRP3) inflammasome [11]. A recent study also found that FGF21 treatment reduced the aortic sinus plaque area and ameliorated dyslipidemia in ApoE^−/−^ mice [10]. Several mechanisms have been suggested for this effect such as improvements in mitochondrial function, decreased oxidative stress, and a reduction in NLRP3-related pyroptosis [13]. Kharitonenkov et al. reported that administration of FGF21 reduced plasma levels of glucose and triglycerides and improved insulin sensitivity in both *ob/ob* and *db/db* mice [14]. Another study on nonhuman primates showed that FGF21 administration decreased the circulating levels of cardiovascular risk markers [15].

In our study, FGF21 treatment reduced fat accumulation in the liver. FGF21 treatment alleviated the fatty liver disease in a mouse model of obesity [16,17] in line with its actions on hepatic lipid oxidation and lipolysis in white adipose tissue (WAT). Conversely, *FGF21* gene knockout mice fed a ketogenic diet exhibited mild obesity and increased hepatic fat accumulation [18]. In diet-induced obesity (DIO) mouse mode, FGF21 treatment inhibited the expression of hepatic sterol regulatory element-binding protein-1 (SREBP-1) gene and a wide array of genes involved in fatty acid and triglyceride synthesis, which was associated with the significant reduction in hepatic triglyceride levels [17]. FGF21 also suppressed cholesterol biosynthesis and attenuated hypercholesterolemia by inhibiting the production of SREBP-2 in hepatocytes [6]. Furthermore, it has been reported that FGF21 prevented atherosclerosis by suppression of the production of hepatic SREBP-2 and induction of adiponectin synthesis in mice [6]

In our study, FGF21 treatment increased HDL-cholesterol and adiponectin levels, and decreased adipocyte size in abdominal visceral adipose tissue. In addition, this treatment reduced the immunopositivity level of CD68 significantly in the abdominal visceral adipose tissue. Adipose tissue is a key target tissue for the action of FGF21, where it is reported to stimulate adiponectin release. This in turn acts on the liver to improve multiple metabolic parameters [19]. One study uncovered the protective effects of FGF21 against atherosclerosis via the induction of adiponectin in adipose tissue, and reduction of hypercholesterolemia by suppression of hepatic SREBP-2 levels [6]. A clinical trial in patients with obesity and DM showed that chronic administration of a long-acting form of FGF21 caused a marked elevation of circulating adiponectin levels and an obvious reduction in blood levels of total and LDL-cholesterol [7]. Consistent with these data, FGF21 treatment reduced weight gain, particularly in fat, improved lipid profiles, and decreased inflammatory cell infiltration in our study.

FGF21 has also been implicated in the “browning” of WAT as it stimulates the expression of mitochondrial brown fat UCP1 in WAT adipocytes, via enhancement of proliferator-activated receptor gamma coactivator-1 α activity and promotes thermogenesis [20]. A study using the same FGF21 analogue molecule as ours (LY2405319) showed improvements in nonalcoholic steatohepatitis in *ob/ob* mice by enhancing hepatic mitochondrial function [21]. Interestingly, FGF21 treatment also activated a thermogenic transcriptional program in BAT by inducing the production of UCP1 and deiodinase-2 [22]. In our study, FGF21 treatment increased the immunopositivity levels of UCP1 compared with controls in the abdominal subcutaneous adipose tissue. We also found that adipocytes in the BAT of FGF21-treated mice were smaller than those in the control group, suggesting activation of BAT from the aspect of energy metabolism. Thus, FGF21 is a potent inducer of UCP1 in WAT, in the browning of WAT, and in the recruitment of thermogenic pathways in BAT, underlying the FGF21-mediated weight loss and improvements in glucose homeostasis.

Interestingly, FGF21 treatment increased total energy expenditure and physical activity levels, inducing body weight loss in a study on DIO mice [17]. The finding of smaller weight gain without difference in the food intake in our study supports the positive effect of FGF21 on energy expenditure. Thus, FGF21 appears to exert its protection against atherosclerosis by multiple actions on the blood vessels, liver, adipose tissues, and exergy metabolism [23].

In the present study, we measured various biochemical markers, such as glucagon, MCP-1, resistin, and β-hydroxybutyrate, which were not measured in previous studies. A recent study has shown that FGF21 promotes ketone body utilization through activation of AMP-dependent kinase [24]. Another study showed that canagliflozin, an SGLT-2 inhibitor, increased fatty acid oxidation, reduced hepatic steatosis, and increased plasma levels of FGF21 [25]. The switch of preferred substrate from glucose to ketone bodies by SGLT-2 inhibitors is of particular interest because ketone bodies can cross the blood–brain barrier and are the normal metabolic substrate for neurons [24]. Given that these phenotypic changes mirror the effects of FGF21, FGF21 would be a good agent to be used with SGLT-2 inhibitors [26].

We confirmed that circulating FGF21 levels were significantly increased by FGF21 analogue administration in our study. Interestingly, it has been reported that serum FGF21 levels were high in people with obesity or metabolic syndrome [27,28]. Resistance to FGF21 might be involved in this phenomenon. By contrast, a sophisticated study in animals has proven that endogenous FGF21 acts as a master regulator to protect against DIO in the absence of UCP1 [29]. Thus, more studies are needed to identify the roles of FGF21 from a cardiometabolic perspective.

There were several limitations to our study. The duration of treatment was not sufficient to identify any long-term effects of FGF21. Moreover, we used the HOMA-IR to evaluate insulin resistance instead of more sophisticated methods. Although it was not validated in animal studies, the HOMA-IR measures has been frequently used in studies using rodent models [30,31]. An investigation of the expression of the FGF receptor/β-Klotho and its downstream pathway could have given more information about the potential role of FGF21 therapy because FGF21 exerts its metabolic actions by binding to the receptors of these factors [6,13]. But we could not investigate the expression of the FGF receptor/β-Klotho and its downstream pathway and the infiltration of macrophages in the plaque due to a lack of available tissues.

In this study, FGF21 analogue therapy was effective against the development of atherosclerosis in addition to its known glucose-lowering property. FGF21 treatment improved insulin sensitivity, estimated by the IPGTT, and increased in adiponectin levels. In addition, reduction in fat accumulation in the liver as well as decrease in fat cell size, less infiltration in inflammatory cells estimated by CD68 immunopositivity, and fewer crown-like structures in visceral adipose tissues can be attributed to the anti-atherosclerotic property of FGF21. It was also found that FGF21 therapy induced UCP1 expression in WAT, in the browning of the WAT, and this is likely to contribute to the FGF21-mediated weight loss and to improvements in glucose homeostasis. We believe that our study adds evidence supporting the potential role of FGF21 in antiatherosclerosis.

## 4. Materials and Methods

### 4.1. Animals

As a mouse model of atherosclerosis, 12-week-old male C57BL/6J ApoE^−/−^ mice were obtained from Jackson Laboratory (Bar Harbor, ME, USA). ApoE^−/−^ mice (*n* = 20) were fed an atherogenic diet containing 1.5 g cholesterol (43% energy from carbohydrate and 41% energy from fat; D12079B; Research Diets, New Brunswick, NJ, USA) for 14 weeks. Then, they were assigned randomly either to control or FGF21 groups. After being fed regular chow for 3 weeks, each group received intraperitoneal injections of the same volume of normal saline or the FGF21 analogue LY2405319 (0.1 mg/kg; hitherto simply FGF21), respectively, for 5 weeks. The study design is shown in Appendix A. The dose of FGF21 was based on a previous study showing beneficial effects on glucose and lipid metabolism [6].

The animals were kept under 12 h/12 h light/dark cycles with free access to food and water. During the lead-in period, one mouse in the FGF21 group died without any obvious cause.

This study was approved by the Institutional Animal Care Committee, Seoul National University Bundang Hospital (SNUBH) (BA1710-233/083-01) (12 October 2017). Animal experiments were performed in compliance with the Guide for Experimental Animal Research of the Laboratory for Experimental Animal Research, Clinical Research Institute, SNUBH, South Korea.

### 4.2. Weight and Food Consumption

Animals were weighed twice a week at the same time in the morning. They were all allowed free access to water and regular chow diet (Purina Korea, Seoul, South Korea), and diet and water consumption were recorded once a week.

### 4.3. IPGTT

Mice were subjected to an IPGTT after 5 weeks of treatment. Each animal was injected intraperitoneally with 1.5 g/kg of a 50 mol/L glucose solution. Blood samples (about 5 μL) were collected from an incision in the tail at baseline and 30, 60, 90 and 120 min after the glucose load. The area under the curve for blood glucose concentration changes with time (AUC_glucose_) was calculated from 0 min to 120 min using the trapezoid rule for glucose data.

### 4.4. Euthanasia and Harvest of Organs

After 5 weeks of treatment with the FGF21 analogue LY2405319 or saline, mice were euthanized with a combination of anesthetics (ketamine, 70 mg/kg IP; xylazine, 7 mg/kg IP; Yuhan Corp, Seoul, South Korea), and the aorta, liver, abdominal subcutaneous and visceral adipose tissues, and BATs were excised rapidly.

### 4.5. Aortic Atherosclerosis in ApoE^−/−^ Mice

To measure the area of atherosclerotic lesions, aortic arches prepared using the *en face* method were stained with Oil Red O solution [32]. After perfusion, whole aorta was dissected out, opened longitudinally from heart to the iliac arteries, pinned on a black wax pan, and stained with Oil red O. To further investigate the existence of plaques in the aortic sinus area, 10 μm-thick cryosections from the mid portion of the ventricle to the aortic arch were obtained after the heart and proximal aorta had been removed. Then, cryosections of the aortic sinus were stained with Oil Red O and hematoxylin. Each section was investigated in a blinded fashion. The images of the aorta were analyzed with the Image-Pro plus program (Media Cybernetics, Silver Spring, MD, USA), and presented as the percentage of lesion area in the whole aorta.

### 4.6. Biochemical Markers Associated with Cardiovascular Risk

At the end of the study period, blood samples were procured after the mice had been fasted for 8 h and biochemical variables were measured using standard methods. Briefly, plasma glucose (glucose oxidase method by YSI 2300-STAT; Yellow Springs Instruments, Yellow Springs, OH, USA) and insulin levels were measured, and homeostatic model assessment of insulin resistance (HOMA-IR) values were calculated [33]. Lipids including triglycerides, high density lipoprotein (HDL)-cholesterol and low density lipoprotein (LDL)-cholesterol, very low density lipoprotein (VLDL), tumor necrosis factor-alpha (TNF-α), interleukin-6 (IL-6), and monocyte chemoattractant protein-1 (MCP-1) were also measured (Multiplex Assay kit [RADPK-81K], Millipore, Billerica, MA, USA), as were adiponectin concentrations (ELISA kits; Millipore). FGF21 and resistin levels were measured using mouse Quantikine ELISA kits (R&D Systems; Minneapolis, MN, USA). β-hydroxybutyrate was measured using colorimetric assay kits (ab83390; Abcam, Cambridge, MA, USA) according to the manufacturer’s instructions.

### 4.7. Histology of Liver and Adipose Tissues and Immunologic Staining for CD68 and UCP1 in the Abdominal Adipose Tissue

The areas and size of lipid droplets that had accumulated in the liver and adipose tissues were measured using light microscope and image analysis software for quantification (Image J software v. 1.50i; National Institutes of Health, Bethesda, MA, USA; https://imagej.nih.gov/ij/download.html) [34]. Immunofluorescence staining of CD68 in the abdominal visceral adipose tissue was performed using an anti-CD68 antibody (1:200) (Abcam). Texas Red X-conjugated goat anti-mouse IgG and Alexa 488-conjugated goat anti-rabbit IgG (1:500) (Invitrogen, Grand Island, NY, USA) were used as secondary antibodies. Sections were mounted and images acquired using fluorescence microscopy (IX81, Olympus, Tokyo, Japan). For immunologic staining of UCP1, abdominal subcutaneous fat was fixed with formalin and paraffin wax-embedded for immunohistochemistry. Sections were immunostained with an anti-UCP1 antibody (1:40,000) (Abcam) and 3,3′-diaminobenzidine (DAB).

### 4.8. Statistical Analysis

The number of mice to be used for the experiment was calculated based on previous studies [32,35]. We anticipated that the mean change in the carotid intima–media ratio, one indicator of atherosclerosis, would be 0.5 in the control group. When two-sided tests were performed at 5% significance level and 80% power and the same number of experimental animals were assigned to each group, the valid number of individuals in each group was predicted to be 10. In addition, considering the possible dropout rate (20%) caused by incidental death, the final population was calculated to be 10 for each group.

Results are reported as the mean ± SD. Analysis of variance was used to test mean differences between groups. Spearman’s test was used for analyzing any correlations. For all tests, *p* < 0.05 was considered to be statistically significant. Analysis was done using IBM SPSS Statistics for Windows (v. 18.0; IBM Corp., Armonk, NY, USA).

## 5. Conclusions

In conclusion, these data suggest that FGF21 molecules such as LY2405319 might be candidate drugs for the treatment of atherosclerosis in patients with DM and obesity. This is of great clinical significance, given that reduction in cardiovascular morbidity and mortality is the ultimate goal in the management of these patients. This study can also be of great help to follow-up studies aiming to evaluate the effect of FGF21 on atherosclerosis, the associated mechanisms, and the effects of combination with other novel antidiabetic medications, such as GLP-1 analogues and SGLT-2 inhibitors.

## Figures and Tables

**Figure 1 ijms-21-06836-f001:**
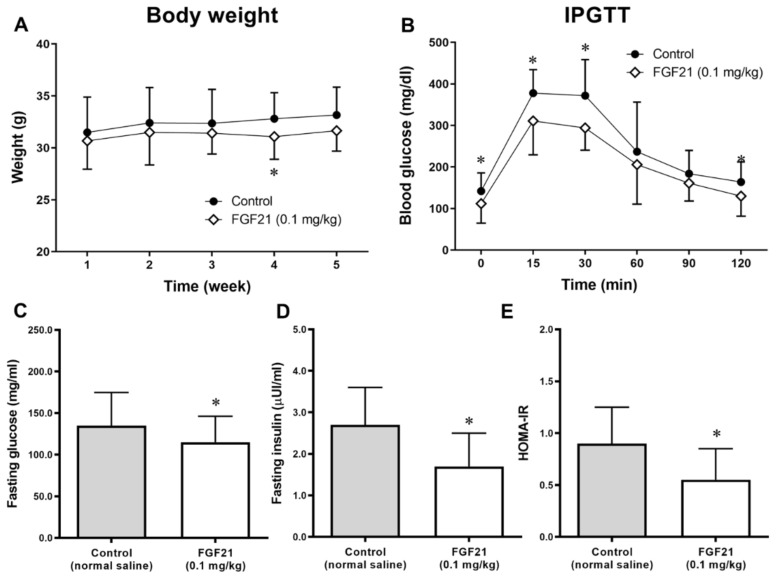
(**A**) Weight changes during the 5 weeks of treatment. (**B**) Glucose levels measured at 0–120 min during the intraperitoneal glucose tolerance test (IPGTT) at the end of study. (**C**–**E**) Fasting glucose and insulin concentrations after 5 weeks of treatment with normal saline or the fibroblast growth factor 21 (FGF21) analogue LY2405319. (**E**) Homeostasis model assessment of insulin resistance (HOMA-IR) calculated from fasting glucose and insulin levels. Data are means ± standard deviation (SD). * *p* < 0.05 vs. control.

**Figure 2 ijms-21-06836-f002:**
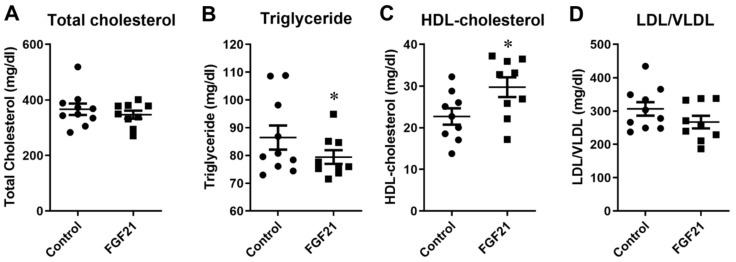
Changes in lipid profiles after 5 weeks of treatment with normal saline as control and fibroblast growth factor 21 (FGF21; 0.1 mg/kg) in the ApoE^−/−^ mice. (**A**). total cholesterol, (**B**). triglyceride, (**C**). HDL-cholesterol, (**D**). LDL/VLDL. * *p* < 0.05 vs. control.

**Figure 3 ijms-21-06836-f003:**
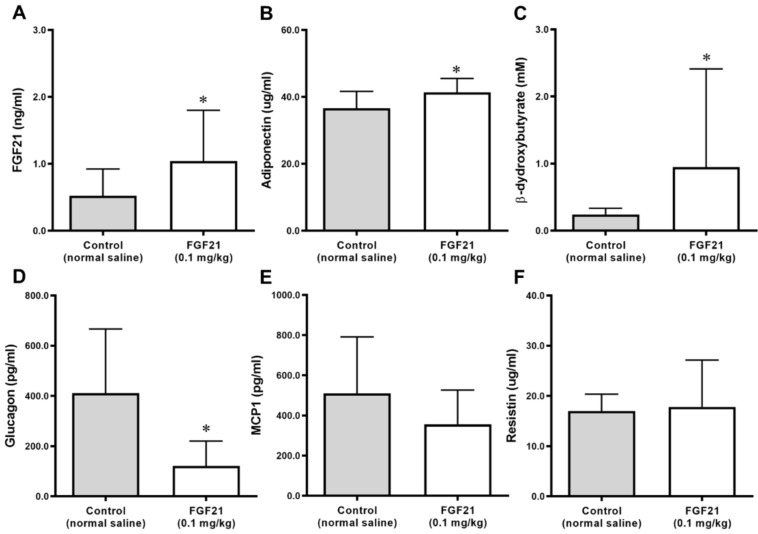
Changes in FGF21 (**A**), adiponectin (**B**), β-hydroxybutyrate (**C**), glucagon (**D**), monocyte chemoattractant protein-1 (MCP1) (**E**), and resistin (**F**) levels after 5 weeks of treatment with normal saline as a control and fibroblast growth factor 21 (FGF21; 0.1 mg/kg) in ApoE^−/−^ mice. Data are means ± SD. * *p* < 0.05 vs. control.

**Figure 4 ijms-21-06836-f004:**
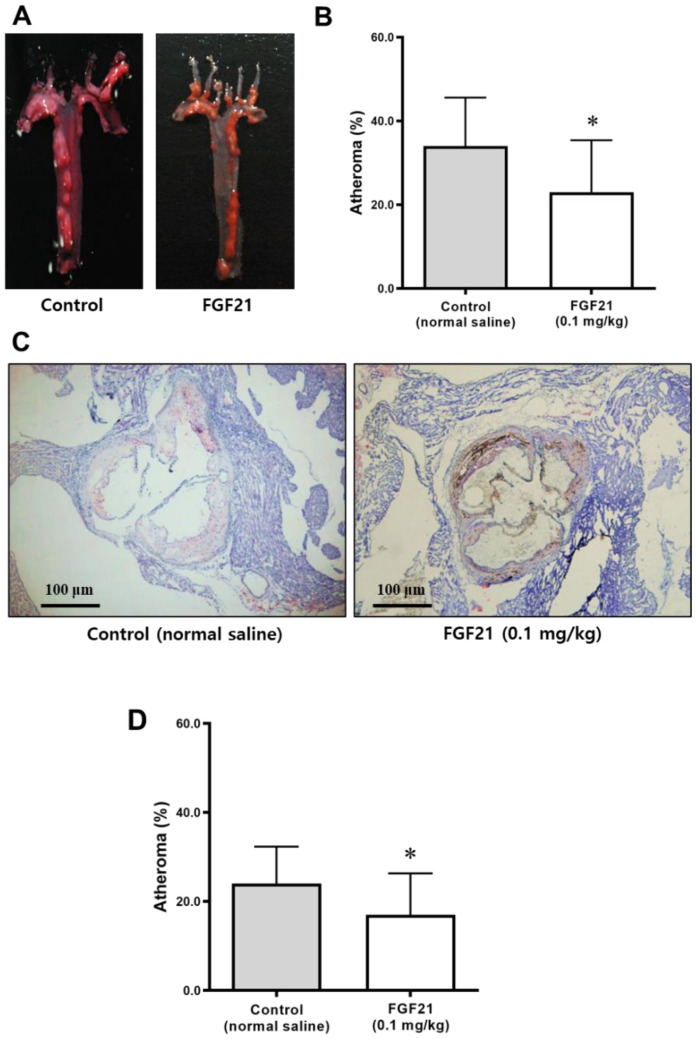
Atheroma burden in the aorta of ApoE^−/−^ mice after 5 weeks of treatment with FGF21 or normal saline. (**A**,**C**) Representative images of atheroma stained using Oil Red O of the aortic arch (**A**) and aortic valve areas (scale bars, 100 μm) (**C**) from ApoE^−/−^ mice. The red color indicates plaque accumulation. (**B**,**D**) The atheromatous plaque area (% of total area) was smaller in the aortic arch (**B**) and aortic valve (**D**) of the FGF21-treated mice compared with the control. Data are means ± SD. Weight-adjusted comparison was used. * *p* < 0.05 vs. control.

**Figure 5 ijms-21-06836-f005:**
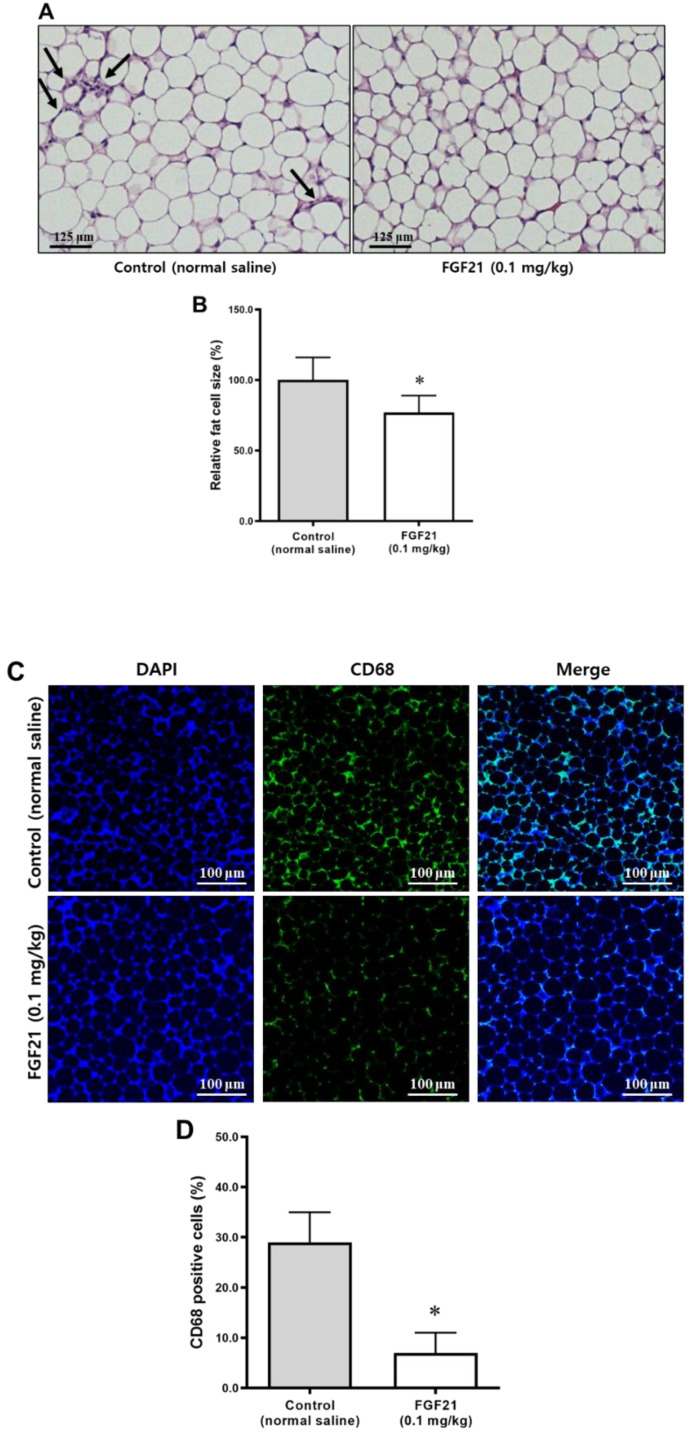
(**A**) Histology of abdominal visceral adipose tissues stained with H&E (scale bars, 20 μm). The arrows indicate crown-like structures. (**B**) Comparison of fat cell size (% of control) between the control and the FGF21 groups. (**C**) Immunohistochemical staining of CD68 in the abdominal visceral adipose tissue (scale bars, 20 μm). (**D**) CD68-positive cells were significantly less abundant in the FGF21-treated mice than in the controls (% of total cells). Data are means ± SD. * *p* < 0.05 vs. control.

**Figure 6 ijms-21-06836-f006:**
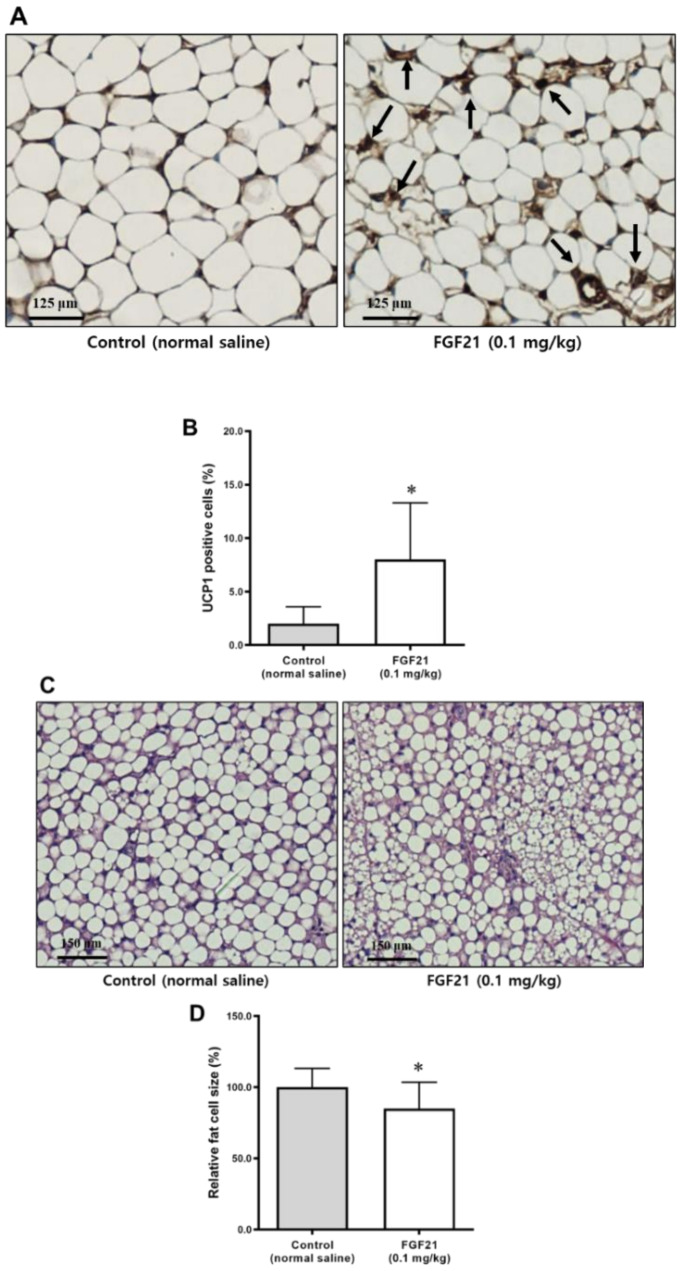
(**A**) Histology of abdominal subcutaneous adipose tissues stained with H&E (scale bars, 20 μm). Arrows indicate uncoupling protein 1 (UCP1)-positive cells. (**B**) UCP1-positive cells were significantly more abundant in the FGF21-treated mice than in the control (% of total cells). (**C**) Histology of brown adipose tissues stained by H&E (scale bars, 20 μm). (**D**) Comparison of fat cell size (% of control) between the control and the FGF21 group. Data are means ± SD. * *p* < 0.05 vs. control.

**Figure 7 ijms-21-06836-f007:**
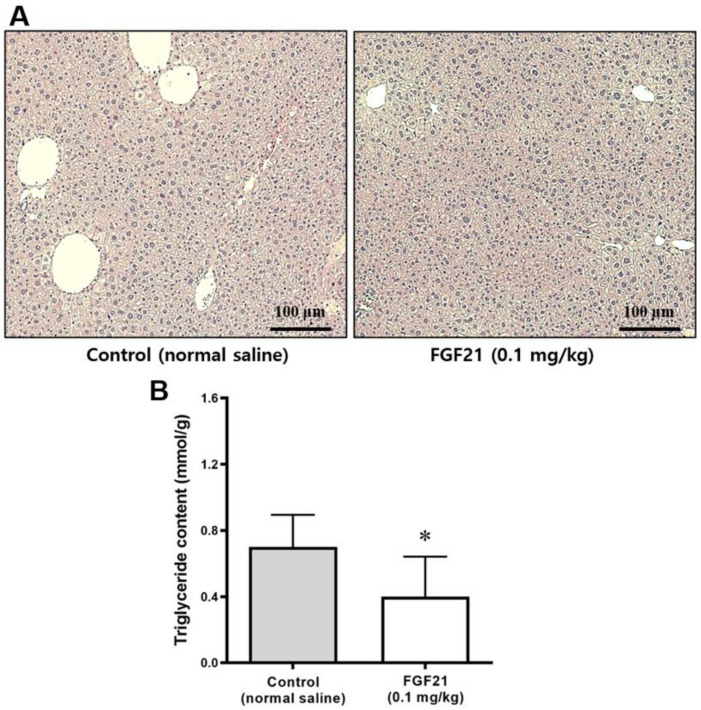
(**A**) Liver histology revealed by H&E staining. All images were acquired at magnification x 200 (*n* = 10 in the control, *n* = 9 in the FGF21 group). (**B**) Triglyceride content in the liver (mmol/g). Data are means ± SD. * *p* < 0.05 vs. control.

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
