# Peer review of "Effect of Fibroblast Growth Factor 21 on the Development of Atheromatous Plaque and Lipid Metabolic Profiles in an Atherosclerosis-Prone Mouse Model"

_ijms, 2020, doi:10.3390/ijms21186836_

Round 1

Reviewer 1 Report

In the manuscriptby Maeng et al, the authors address atheroprotective effects of FGF21 by injecting ApoE-/- mice with FGF21 analoge, LY2405319. The paper adresses improtant issue. However, there are some issues that must be addressed by the authors. The paper is somewhat descriptive and offers no new insight on the atheroprotective role of FGF21.

1. The authors should mention, what was the background strain of their ApoE-/- mice because mice of different background differ in their insulin secretion and/or resistance.

2. The authors refer to the paper by Lin et al (Circulation 2015, 131, (21), 1861-71) for the selection of LY240539 dose. In the paper by Lin et al, recombinant FGF21 was used. In addition, double knockout ApoE-/- and FGF21-/- mice were used. In the present manuscript by Maeng et al the effects of LY2405319 are minor and often do not reach statistical significance level. Can this be due to not optimal LY2405319 dosage in ApoE-/- mice or FGF21 resistance? Expression of FGFR/b-Klotho or activation of downstream pathways could be analyzed.

3. In the methods authors state thate “data are shown as mean +/- SD or SEM as indicated”. In Fig. 1 it is not indicated, in Figs 2-5 SEM is shown. The data in all figures should be be presented as SD and not SEM (see PMID: 16223828)

4. Figure 1C. Better images should be provided, especially for FGF21-treatment – no valves are visible. More details should be provided in methods such as distance from the aortic sinus should be given. Infiltration of macrophages in the plaques should be assessed.

5. In the title and in the abstract the authors state that atherosclerosys pathways were assessed in the manuscript. However, atherosclerosis-related data (HDL, LDL/VLDL) are mainly shown in supplement.

6. Inflammatory properties of adipose tissue should be analyzed.

7. In the lines 163-164 the authors state that “only few studies have been conducted to investigate role of FGF21 in atherosclerosis”. There are more studies available from Pubmed that should be mentioned (to name a few PMID: 30157856; PMID: 32445748; PMID: 31513948

Author Response

[Response to Reviewer 1]

In the manuscript by Maeng et al, the authors address atheroprotective effects of FGF21 by injecting ApoE-/- mice with FGF21 analoge, LY2405319. The paper adresses improtant issue. However, there are some issues that must be addressed by the authors. The paper is somewhat descriptive and offers no new insight on the atheroprotective role of FGF21.

  1. The authors should mention, what was the background strain of their ApoE-/- mice because mice of different background differ in their insulin secretion and/or resistance.

As noted in Materials and Methods, the background of ApoE/ mice used in this study was C57BL/6J (Jackson Laboratory; Bar Harbor, ME, USA).

[Revised] p. 15, lines 315316 in the revised manuscript.

As a mouse model of atherosclerosis, 12-week-old male C57BL/6J ApoE−/− mice were obtained from Jackson Laboratory (Bar Harbor, ME, USA).

  1. The authors refer to the paper by Lin et al (Circulation 2015, 131, (21), 1861-71) for the selection of LY240539 dose. In the paper by Lin et al, recombinant FGF21 was used. In addition, double knockout ApoE-/- and FGF21-/- mice were used. In the present manuscript by Maeng et al the effects of LY2405319 are minor and often do not reach statistical significance level. Can this be due to not optimal LY2405319 dosage in ApoE-/- mice or FGF21 resistance? Expression of FGFR/b-Klotho or activation of downstream pathways could be analyzed.

 We thank the reviewer for this helpful comment. There could be several reasons why the treatment of LY2405319 did not reach statistical significance in some experiments. As the reviewer suggests, the dose of LY2405319 used in our study might not have been enough to show its efficacy significantly. The effective dose of some medications could differ according to the study species (e.g., humans vs. animals and mice vs. rats). Several studies have reported distinct differences in the physiological regulation and functions of FGF21 among mammalian species, such as changes in its expression by fasting and refeeding or by growth hormone treatment [1-3]. In addition, there have been several studies suggesting FGF21 resistance. Despite the multiple favorable metabolic effects of FGF21, circulating FGF21 was found to be elevated in humans with obesity and in diet-induced obese rodents [4, 5], which suggests the existence of resistance to FGF21, particularly in obese conditions [3]. In addition, it would be informative to investigate expression of the FGF receptor/b-Klotho or activation of downstream pathways [3, 6]. However, there were not enough tissues to perform this experiment. Instead, we have mentioned this as a limitation of our study.

[Revised] p. 14–15, lines 289–294 in the revised manuscript.

We confirmed that circulating FGF21 levels were significantly increased by FGF21 analogue administration in our study. Interestingly, it has been reported that serum FGF21 levels were high in people with obesity or metabolic syndrome [27, 28]. Resistance to FGF21 might be involved in this phenomenon. By contrast, a sophisticated study in animals has proven that endogenous FGF21 acts as a master regulator to protect against DIO in the absence of UCP1 [29]. Thus, more studies are needed to identify the roles of FGF21 from a cardiometabolic perspective.

[Revised] p. 15, lines 298303 in the revised manuscript.

An investigation of the expression of the FGF receptor/b-Klotho and its downstream pathway could have given more information about the potential role of FGF21 therapy because FGF21 exerts its metabolic actions by binding to the receptors of these factors [6, 13]. But we could not investigate the expression of the FGF receptor/b-Klotho and its downstream pathway and the infiltration of macrophages in the plaque due to a lack of available tissues.

[References for revision]

  1. Chen, W.; Hoo, R. L.; Konishi, M.; Itoh, N.; Lee, P. C.; Ye, H. Y.; Lam, K. S.; Xu, A., Growth hormone induces hepatic production of fibroblast growth factor 21 through a mechanism dependent on lipolysis in adipocytes. J Biol Chem 2011, 286, (40), 34559-66.
  2. Lundberg, J.; Hoybye, C.; Krusenstjerna-Hafstrom, T.; Bina, H. A.; Kharitonenkov, A.; Angelin, B.; Rudling, M., Influence of growth hormone on circulating fibroblast growth factor 21 levels in humans. J Intern Med 2013, 274, (3), 227-32.
  3. Geng, L.; Lam, K. S. L.; Xu, A., The therapeutic potential of FGF21 in metabolic diseases: from bench to clinic. Nat Rev Endocrinol 2020.
  4. Zhang, X.; Yeung, D. C.; Karpisek, M.; Stejskal, D.; Zhou, Z. G.; Liu, F.; Wong, R. L.; Chow, W. S.; Tso, A. W.; Lam, K. S.; Xu, A., Serum FGF21 levels are increased in obesity and are independently associated with the metabolic syndrome in humans. Diabetes 2008, 57, (5), 1246-53.
  5. Fisher, F. M.; Chui, P. C.; Antonellis, P. J.; Bina, H. A.; Kharitonenkov, A.; Flier, J. S.; Maratos-Flier, E., Obesity is a fibroblast growth factor 21 (FGF21)-resistant state. Diabetes 2010, 59, (11), 2781-9.
  6. Lin, Z.; Pan, X.; Wu, F.; Ye, D.; Zhang, Y.; Wang, Y.; Jin, L.; Lian, Q.; Huang, Y.; Ding, H.; Triggle, C.; Wang, K.; Li, X.; Xu, A., Fibroblast growth factor 21 prevents atherosclerosis by suppression of hepatic sterol regulatory element-binding protein-2 and induction of adiponectin in mice. Circulation 2015, 131, (21), 1861-71.

  1. In the methods authors state thate “data are shown as mean +/- SD or SEM as indicated”. In Fig. 1 it is not indicated, in Figs 2-5 SEM is shown. The data in all figures should be be presented as SD and not SEM (see PMID: 16223828)

 We thank the reviewer for this insightful comment. As suggested, we have presented the data with their means ± SDs in all figures.

  1. Figure 1C. Better images should be provided, especially for FGF21-treatment – no valves are visible. More details should be provided in methods such as distance from the aortic sinus should be given. Infiltration of macrophages in the plaques should be assessed.

 We have provided a new image of the aortic valve area from the FGF21 treatment group in the revised manuscript (now Figure 4). As suggested, more detailed information regarding this method has been provided in the methodology. Unfortunately, sufficient tissue was not available to investigate the infiltration of macrophages in the plaques. We have added this information as a limitation of this study.

[Revised Figure 4] please check revised Figure 4 in manuscript.

[Revised] p. 17, lines 349352 in the revised manuscript.

To further investigate the existence of plaques in the aortic sinus area, 10 μm-thick cryosections from the mid portion of the ventricle to the aortic arch were obtained after the heart and proximal aorta had been removed. Then, cryosections of the aortic sinus were stained with Oil Red O and hematoxylin. Each section was investigated in a blinded fashion.

[Revised] p. 15, lines 298303 in the revised manuscript.

An investigation of the expression of the FGF receptor/b-Klotho and its downstream pathway could have given more information about the potential role of FGF21 therapy because FGF21 exerts its metabolic actions by binding to the receptors of these factors [6, 13]. But we could not investigate the expression of the FGF receptor/b-Klotho and its downstream pathway and the infiltration of macrophages in the plaque due to a lack of available tissues.

  1. In the title and in the abstract the authors state that atherosclerosys pathways were assessed in the manuscript. However, atherosclerosis-related data (HDL, LDL/VLDL) are mainly shown in supplement.

 According to the reviewer’s suggestion, we have changed the Title and the content in the abstract to “development of atherosclerosis and its associated parameters.” in the revised manuscript.

[Revised title]

Effect of Fibroblast Growth Factor 21 on the Development of Atheromatous Plaque and Lipid Metabolic Profiles in an Atherosclerosis-Prone Mouse Model

[Revised] p. 1, lines 1516 in the revised manuscript.

We aimed to investigate the effect of an FGF21 analogue (LY2405319) on the development of atherosclerosis and its associated parameters.

  1. Inflammatory properties of adipose tissue should be analyzed.

 In this study, we assessed crown-like structures in adipose tissue that indicate the infiltration of inflammatory cells. These structures were found more frequently in the control group than in the FGF21-treated group (Figure 3A). FGF21 treatment also significantly reduced the immunopositivity level of CD68 in abdominal visceral adipose tissue. CD68 is associated with the adipose tissue macrophage burden [7, 8]. In addition, FGF21 treatment increased adiponectin levels in our experiment. It is well known that adiponectin has anti-inflammatory properties [9-11]. Based on these findings, we suggest that FGF21 treatment might reduce inflammation in multiple ways.

[Revised] p. 13, lines 251262 in the revised manuscript.

In our study, FGF21 treatment increased HDL-cholesterol and adiponectin levels, and decreased adipocyte size in abdominal visceral adipose tissue. In addition, this treatment reduced the immunopositivity level of CD68 significantly in the abdominal visceral adipose tissue. Adipose tissue is a key target tissue for the action of FGF21, where it is reported to stimulate adiponectin release. This in turn acts on the liver to improve multiple metabolic parameters [19]. One study uncovered the protective effects of FGF21 against atherosclerosis via the induction of adiponectin in adipose tissue, and reduction of hypercholesterolemia by suppression of hepatic SREBP-2 levels [6]. A clinical trial in patients with obesity and DM showed that chronic administration of a long-acting form of FGF21 caused a marked elevation of circulating adiponectin levels and an obvious reduction in blood levels of total and LDL-cholesterol [7]. Consistent with these data, FGF21 treatment reduced weight gain, particularly in fat, improved lipid profiles, and decreased inflammatory cell infiltration in our study.

[Revised] p. 15, lines 306309 in the revised manuscript.

FGF21 treatment improved insulin sensitivity, estimated by the IPGTT, and increased in adiponectin levels. In addition, reduction in fat accumulation in the liver as well as decrease in fat cell size, less infiltration in inflammatory cells estimated by CD68 immunopositivity, and fewer crown-like structures in visceral adipose tissues can be attributed to the anti-atherosclerotic property of FGF21.

[References for revision]

  1. Graff, E. C.; Fang, H.; Wanders, D.; Judd, R. L., The Absence of Adiponectin Alters Niacin's Effects on Adipose Tissue Inflammation in Mice. Nutrients 2020, 12, (8).
  2. Jia, Q.; Morgan-Bathke, M. E.; Jensen, M. D., Adipose tissue macrophage burden, systemic inflammation, and insulin resistance. Am J Physiol Endocrinol Metab 2020, 319, (2), E254-E264.
  3. Choi, H. M.; Doss, H. M.; Kim, K. S., Multifaceted Physiological Roles of Adiponectin in Inflammation and Diseases. Int J Mol Sci 2020, 21, (4).
  4. Cersosimo, E.; Xu, X.; Terasawa, T.; Dong, L. Q., Anti-inflammatory and anti-proliferative action of adiponectin mediated by insulin signaling cascade in human vascular smooth muscle cells. Mol Biol Rep 2020.
  5. Maeda, N.; Funahashi, T.; Matsuzawa, Y.; Shimomura, I., Adiponectin, a unique adipocyte-derived factor beyond hormones. Atherosclerosis 2020, 292, 1-9.

  1. In the lines 163-164 the authors state that “only few studies have been conducted to investigate role of FGF21 in atherosclerosis”. There are more studies available from Pubmed that should be mentioned (to name a few PMID: 30157856; PMID: 32445748; PMID: 31513948

As the reviewer suggests, we have included these references and have discussed their results and implications in the revised manuscript.

[Revised] p. 12, lines 228236 in the revised manuscript.

Several studies have been conducted to investigate the role of FGF21 in atherosclerosis [6, 7, 10-12]. A study using ApoE–/– mice reported that the administration of FGF21 alleviated atherosclerosis by ameliorating Fas-mediated apoptosis [12]. Another study demonstrated the protective effect of FGF21 on the proliferation and migration of vascular smooth muscle cells via inhibition of the nucleotide-binding domain leucine-rich repeat and pyrin domain containing receptor 3 (NLRP3) inflammasome [11]. A recent study also found that FGF21 treatment reduced the aortic sinus plaque area and ameliorated dyslipidemia in ApoE–/– mice [10]. Several mechanisms have been suggested for this effect such as improvements in mitochondrial function, decreased oxidative stress, and a reduction in NLRP3-related pyroptosis [13].

Reviewer 2 Report

In this manuscript, authors applied the mouse ApoE-/-mouse model with atherogenic diet to evaluate the efficacy of FGF21 analogue in ameliorating atherosclerosis and regulating metabolic profiles. Although most findings in this manuscript have been reported in recent years, this study remains to provide certain meaningful information to readers. However, the few data limit the importance of this study. In addition, there are few points needed to be clarified.

  1. The application of FGF21 for treating atherosclerosis in ApoE-/- mice has been reported recently (Lipids Health Dis. 2018; 17: 203). Authors should cite this article and add a paragraph to compare findings of these two studies in the Discussion section.
  2. Although most findings presented in the manuscript have been reported in several separated studies, authors should make a connection to link all experimental results in this manuscript to augment the importance and novelty of their work. Moreover, Figure S3 and S4 should be moved to the Results section, but not in the supplemental part.
  3. What’s the tissue used for immunostaining against CD68? The figure legend (Figure 3, line 121) indicates CD68 signals were detected in the abdominal visceral adipose tissue, while in the Materials and Methods section (line 269), the immunostaining of CD68 was indicated for detecting infiltration of macrophages (CD68+ cells) in the atheroma. Moreover, the image quality of Fig. 3C should be improved, and the quantitation of CD68+ cell (Fig. 3D) in not matched the image of immunostaining.
  4. The words “cardiometabolic profiles” in title is suggested to be removed or replaced with “lipid metabolic”.
  5. The section 4.7, 4.8, and 4.9 should be merged to as a single paragraph.

Author Response

[Response to Reviewer 2]

In this manuscript, authors applied the mouse ApoE-/-mouse model with atherogenic diet to evaluate the efficacy of FGF21 analogue in ameliorating atherosclerosis and regulating metabolic profiles. Although most findings in this manuscript have been reported in recent years, this study remains to provide certain meaningful information to readers. However, the few data limit the importance of this study. In addition, there are few points needed to be clarified.

  1. The application of FGF21 for treating atherosclerosis in ApoE-/- mice has been reported recently (Lipids Health Dis. 2018; 17: 203). Authors should cite this article and add a paragraph to compare findings of these two studies in the Discussion section.

We have included the references suggested by reviewers #1 and #2 and have discussed these in the revised manuscript.

[Revised] p. 12, lines 228236 in the revised manuscript.

Several studies have been conducted to investigate the role of FGF21 in atherosclerosis [6, 7, 10-12]. A study using ApoE–/– mice reported that the administration of FGF21 alleviated atherosclerosis by ameliorating Fas-mediated apoptosis [12]. Another study demonstrated the protective effect of FGF21 on the proliferation and migration of vascular smooth muscle cells via inhibition of the nucleotide-binding domain leucine-rich repeat and pyrin domain containing receptor 3 (NLRP3) inflammasome [11]. A recent study also found that FGF21 treatment reduced the aortic sinus plaque area and ameliorated dyslipidemia in ApoE–/– mice [10]. Several mechanisms have been suggested for this effect such as improvements in mitochondrial function, decreased oxidative stress, and a reduction in NLRP3-related pyroptosis [13].

  1. Although most findings presented in the manuscript have been reported in several separated studies, authors should make a connection to link all experimental results in this manuscript to augment the importance and novelty of their work. Moreover, Figure S3 and S4 should be moved to the Results section, but not in the supplemental part.

We appreciate this helpful comment. As the reviewer suggests, we have moved Figures S3 and S4 to the Results section in the revised manuscript. We have also integrated our study results to augment the novelty of our experiments.

[Revised] p. 15, lines 304312 in the revised manuscript.

In this study, FGF21 analogue therapy was effective against the development of atherosclerosis in addition to its known glucose-lowering property. FGF21 treatment improved insulin sensitivity, estimated by the IPGTT, and increased in adiponectin levels. In addition, reduction in fat accumulation in the liver as well as decrease in fat cell size, less infiltration in inflammatory cells estimated by CD68 immunopositivity, and fewer crown-like structures in visceral adipose tissues can be attributed to the anti-atherosclerotic property of FGF21. It was also found that FGF21 therapy induced UCP1 expression in WAT, in the browning of the WAT, and this is likely to contribute to the FGF21-mediated weight loss and to improvements in glucose homeostasis. We believe that our study adds evidence supporting the potential role of FGF21 in antiatherosclerosis.

  1. What’s the tissue used for immunostaining against CD68? The figure legend (Figure 3, line 121) indicates CD68 signals were detected in the abdominal visceral adipose tissue, while in the Materials and Methods section (line 269), the immunostaining of CD68 was indicated for detecting infiltration of macrophages (CD68+ cells) in the atheroma. Moreover, the image quality of Fig. 3C should be improved, and the quantitation of CD68+ cell (Fig. 3D) in not matched the image of immunostaining.

 Immunostaining against CD68 was performed with the abdominal visceral adipose tissues, not with atheromas in the aortic valve area. We have updated this information to avoid any confusion in the revised manuscript.

Regarding Figure 3C, we have provided a new Figure for this (now Figure 5C) for better image quality. We have also corrected the data for quantitation of CD68+ cells in Figure 5D.

[Revised] p. 18, lines 373374 in the revised manuscript.

Immunofluorescence staining of CD68 in the abdominal visceral adipose tissue was performed using an anti-CD68 antibody (1:200) (Abcam).

[Please see the revised Figure 5]

  1. The words “cardiometabolic profiles” in title is suggested to be removed or replaced with “lipid metabolic”.

We have changed the title as suggested.

[Revised title]

Effect of Fibroblast Growth Factor 21 on the Development of Atheromatous Plaque and Lipid Metabolic Profiles in an Atherosclerosis-Prone Mouse Model

  1. The section 4.7, 4.8, and 4.9 should be merged to as a single paragraph.

 As the reviewer suggests, we have merged these sections into a single paragraph.

[Revised] p. 18, lines 368379 in the revised manuscript.

4.7. Histology of Liver and Adipose Tissues and Immunologic Staining for CD68 and UCP1 in the Abdominal Adipose Tissue

The areas and size of lipid droplets that had accumulated in the liver and adipose tissues were measured using light microscope and image analysis software for quantification (Image J software v. 1.50i; National Institutes of Health, Bethesda, MA, USA; https://imagej.nih.gov/ij/download.html) [34]. Immunofluorescence staining of CD68 in the abdominal visceral adipose tissue was performed using an anti-CD68 antibody (1:200) (Abcam). Texas Red X-conjugated goat anti-mouse IgG and Alexa 488-conjugated goat anti-rabbit IgG (1:500) (Invitrogen, Grand Island, NY, USA) were used as secondary antibodies. Sections were mounted and images acquired using fluorescence microscopy (IX81, Olympus, Tokyo, Japan). For immunologic staining of UCP1, abdominal subcutaneous fat was fixed with formalin and paraffin wax-embedded for immunohistochemistry. Sections were immunostained with an anti-UCP1 antibody (1:40000) (Abcam) and 3,3′-diaminobenzidine (DAB).

Round 2

Reviewer 1 Report

The authors addressed my questions.